# Role of Hibernation Promoting Factor in Ribosomal Protein Stability during *Pseudomonas aeruginosa* Dormancy

**DOI:** 10.3390/ijms21249494

**Published:** 2020-12-14

**Authors:** Sokuntheary Theng, Kerry S. Williamson, Michael J. Franklin

**Affiliations:** 1Department of Microbiology and Immunology, Montana State University, Bozeman, MT 59717, USA; theary_theng@yahoo.com (S.T.); ksw@montana.edu (K.S.W.); 2Center for Biofilm Engineering, Montana State University, Bozeman, MT 59717, USA

**Keywords:** *Pseudomonas aeruginosa*, biofilm, ribosome, ribosomal protein, hibernation promoting factor, stringent response, dormancy

## Abstract

*Pseudomonas aeruginosa* is an opportunistic pathogen that causes biofilm-associated infections. *P. aeruginosa* can survive in a dormant state with reduced metabolic activity in nutrient-limited environments, including the interiors of biofilms. When entering dormancy, the bacteria undergo metabolic remodeling, which includes reduced translation and degradation of cellular proteins. However, a supply of essential macromolecules, such as ribosomes, are protected from degradation during dormancy. The small ribosome-binding proteins, hibernation promoting factor (HPF) and ribosome modulation factor (RMF), inhibit translation by inducing formation of inactive 70S and 100S ribosome monomers and dimers. The inactivated ribosomes are protected from the initial steps in ribosome degradation, including endonuclease cleavage of the ribosomal RNA (rRNA). Here, we characterized the role of HPF in ribosomal protein (rProtein) stability and degradation during *P. aeruginosa* nutrient limitation. We determined the effect of the physiological status of *P. aeruginosa* prior to starvation on its ability to recover from starvation, and on its rRNA and rProtein stability during cell starvation. The results show that the wild-type strain and a stringent response mutant (∆*relA*∆*spoT* strain) maintain high cellular abundances of the rProteins L5 and S13 over the course of eight days of starvation. In contrast, the abundances of L5 and S13 reduce in the ∆*hpf* mutant cells. The loss of rProteins in the ∆*hpf* strain is dependent on the physiology of the cells prior to starvation. The greatest rProtein loss occurs when cells are first cultured to stationary phase prior to starvation, with less rProtein loss in the ∆*hpf* cells that are first cultured to exponential phase or in balanced minimal medium. Regardless of the pre-growth conditions, *P. aeruginosa* recovery from starvation and the integrity of its rRNA are impaired in the absence of HPF. The results indicate that protein remodeling during *P. aeruginosa* starvation includes the degradation of rProteins, and that HPF is essential to prevent rProtein loss in starved *P. aeruginosa*. The results also indicate that HPF is produced throughout cell growth, and that regardless of the cellular physiological status, HPF is required to protect against ribosome loss when the cells subsequently enter starvation phase.

## 1. Introduction

*Pseudomonas aeruginosa* is an opportunistic pathogen that causes chronic infections, including pulmonary infections in patients with the genetic disorder, cystic fibrosis (CF) [1,2]. Chronic infections are often associated with biofilms [3], where bacteria are encased in their extracellular polymers and attached to surfaces [4]. Chronic biofilm infections are difficult to eliminate with antibiotic treatments [5], in part because the biofilms contain cells in a variety of different physiological states, including cells that are dormant [6]. Since most antibiotics only target active metabolic processes, dormant bacteria are tolerant to antibiotics at concentrations that are effective against planktonic cells [7,8,9]. Studies of *P. aeruginosa* isolates from CF patients indicate that even when patients are treated with antibiotics, clones of the original infecting strain reemerge, suggesting that a subpopulation of the founding infections survives the treatment and repopulates the biofilm when treatments are alleviated [10,11,12]. In order to target the dormant bacteria of chronic infections, it will be important to gain a better understanding of the physiology of the dormant subpopulation of infectious biofilms [13].

When bacteria undergo a shift from active growth to dormancy, they undergo a variety of metabolic changes [13]. Included in these metabolic processes is a shift away from energy-expensive processes, such as translation, and conservation of energy stores [13,14]. During starvation, many bacteria also undergo a reduction in cell size and metabolic reprograming, which includes degradation of certain proteins, to be used as energy, carbon, or nitrogen resources. When bacteria are nutrient-limited, they reduce translation and reduce their number of ribosomes. However, a critical number of ribosomes is essential for resuscitation from dormancy, and therefore a supply of ribosomes is preserved in dormant cells [15]. As shown for *Escherichia coli* and many other species of bacteria, ribosomes are inactivated during cell starvation by the small ribosome-binding accessory proteins, ribosome modulation factor (RMF) and ribosome hibernation factor (HPF) [16,17,18,19,20]. In *P. aeruginosa*, HPF is also critical for maintenance of rRNA integrity during starvation and for optimal resuscitation of the cells from starvation [21,22].

Structural studies have examined the roles of RMF and HPF in ribosome binding and inactivation for a variety of bacteria [23,24,25,26,27,28,29,30]. In Gram-negative bacteria, RMF binds to ribosomes in the mRNA channel near the small subunit protein, S1, preventing interaction between the anti-Shine Dalgarno site and the Shine Dalgarno sequence of the mRNA [24,29]. HPF binds near the A-site, inhibiting tRNA binding between the codon and anti-codon [24,29]. Binding of RMF results in ribosome dimerization and the formation of inactive 90S ribosome dimers [31]. HPF binding converts the ribosomes to an inactive 100S dimer form [31,32]. In *E. coli*, an HPF homolog, YfiA, competes with HPF, forming inactive 70S ribosome monomers where the C-terminal tail of YfiA inhibits the binding of RMF to the ribosome [31]. In contrast, Gram-positive bacteria do not have the gene for RMF, but have an HPF with a long C-terminal tail [23,26]. The HPF from Gram-positive bacteria causes the formation of inactive 100S ribosome dimers by protein-protein interactions of its C-terminal domain, caused by formation of HPF homodimers within its binding site of the ribosome [23,26].

HPF binding to ribosomes causes association of the 30S and 50S ribosomal subunits into inactive 70S ribosome monomers and 100S ribosome dimers. Subunit association hinders cleavage of rRNA by endonucleases, preventing the initial steps in ribosome degradation [33,34]. In *Bacillus subtilis*, the absence of HPF results in the loss of rProteins proteins S2 and S3 from the ribosomes of stationary phase cells [35]. Therefore, cleavage of rRNA [34] and loss of rProteins [35] represent the early stages of ribosome recycling. Since HPF is broadly distributed in bacteria and since heterologous HPFs protect ribosome integrity in *P. aeruginosa* [22], HPF likely evolved to protect against these initial steps in ribosome degradation, and ensure a sufficient supply of ribosomes are available when the bacteria emerge from dormancy.

In previous studies, we demonstrated that HPF, but not RMF, is essential for optimal resuscitation of *P. aeruginosa* from dormancy, and for preservation of the rRNA component of ribosomes [21,22]. Here, we further characterized ribosome recycling of *P. aeruginosa* by characterizing preservation and degradation of rProteins during *P. aeruginosa* starvation. We tested the roles of HPF and the stringent response in rProtein preservation. We also tested the effect of the physiological status of bacteria prior to starvation on rProtein and rRNA degradation/preservation. The results show that HPF, but not the stringent response, is required for protection of rProteins from degradation during planktonic starvation. The physiological status of the cells prior to starvation also plays a role in the rate of rProtein recycling, but HPF is required for rRNA integrity during starvation, regardless of the cell’s physiological status.

## 2. Results

**Effect of HPF and the stringent response on ribosomal proteins during *P. aeruginosa* growth.** In order to monitor ribosomal protein abundance during *P. aeruginosa* cell growth and starvation, we induced polyclonal antibody production to the *P. aeruginosa* HPF protein and to the large and small subunit ribosomal proteins, L5 and S13. L5 and S13 were chosen because these proteins interact during active translation, forming the B1b bridge, and are essential for ribosome ratcheting during translation [36,37]. Antibodies to these proteins were generated by fusing the *P. aeruginosa hpf*, *rplE*, and *rpsM* genes to 6X-poly-histidine tags, purifying the proteins, and using the purified proteins to induce rabbit polyclonal antibodies. The IgG fraction was purified from the antisera and tested for efficacy in immunoblot analyses against the *P. aeruginosa* proteins. For HPF, a band near the molecular mass of HPF (11.7 kDa) was observed in the *P. aeruginosa* PAO1 cell extracts, but not in extracts from the *P. aeruginosa* ∆*hpf* mutant strain (Appendix A). Complementation of *P. aeruginosa* ∆*hpf* with *hpf* on a plasmid resulted in high expression of HPF (Appendix A). For the large subunit protein L5, antibodies reacted with a band at approximately 20 kDa, the calculated molecular mass of L5 (Appendix A). Since L5 is essential [38], no deletion mutant was available to serve as a negative control in immunoblots. Therefore, we tested the antibodies against L5 fusion proteins. Antibodies reacted with the L5-6X-His fusion protein, giving a band of 21 kDa, and the L5-YFP (yellow fluorescent protein) fusion protein, giving a band of 48 kDa (Appendix A). Similarly, for the small ribosomal subunit protein S13, a band was observed at approximately the molecular mass of S13 (13.3 kDa) (Appendix A), and the antibodies reacted with an S13-YFP fusion protein, producing a band at 41 kDa. The results indicated that these antibodies are effective for use in immunoblot analysis to detect these ribosomal proteins during *P. aeruginosa* growth and starvation.

We first used the antibodies in immunoblots to determine the relative amounts of HPF, L5, and S13 over the course of *P. aeruginosa* growth. Growth curves were performed over 24 h and assayed for colony forming units (CFUs) (Appendix A), total protein content (Appendix A), and abundances of HPF, L5 and S13 (Appendix A). Growth curves and total cellular protein amounts showed that there was no effect of the ∆*hpf* deletion during *P. aeruginosa* growth in TSB medium. Overexpression of *hpf* also did not influence cell growth. For immunoblot analysis, total cellular protein concentrations were determined and used to load 15 μg cellular protein per lane (Appendix A). For all rProtein abundance studies, band intensities from at least three independent biological replicates were quantified by densitometry. Immunoblots over the course of cell growth showed that HPF amounts were relatively constant throughout exponential phase and into early stationary phase in wild-type *P. aeruginosa* PAO1 (*p* = 0.46) (Appendix A). No HPF bands were observed in the ∆*hpf* strain. *P. aeruginosa* ∆*hpf* containing *hpf* on a plasmid under control of a strong promoter (P*_trc_*-*hpf*) resulted in increased amounts of HPF compared to the wild-type strain (*p* < 0.0001) (Appendix A).

The stringent response is a nutrient starvation response caused by ribosome stalling due to insufficient amino acid-charged tRNAs. The stringent response results from the production of the starvation signaling molecule, guanosine tetraphosphate (or pentaphosphate), (p)ppGpp, through the activities of RelA and SpoT (reviewed in [39]). Since the stringent response is involved in ribosome dimerization in *E. coli* [40], and has a modulatory effect on *hpf* expression in *P. aeruginosa* [41], here, we determined if the stringent response affected HPF levels in *P. aeruginosa* over the course of growth and starvation. *P. aeruginosa* growth and cellular protein abundance over growth was not affected by the ∆*relA*∆*spoT* mutations (Appendix A). In early exponential phase, HPF production was delayed in the ∆*relA*∆*spoT* strain compared to the wild-type strain (Appendix A), similar to the modulatory effect we observed for *hpf* expression in the ∆*relA*∆*spoT* mutant [41]. HPF protein amounts in the ∆*relA*∆*spoT* mutant ultimately reached similar levels to that of the wild-type strain (*p* = 0.50) (Appendix A).

The ribosomal protein L5 showed consistent levels of protein abundance over the course of exponential growth in the wild-type strain, the ∆*hpf* mutant, and the ∆*relA*∆*spoT* mutant, although the L5 protein had reduced abundances at 8 hours and into stationary phase in PAO1 and the ∆*hpf* mutant (Appendix A). The ribosomal protein S13 abundance was not affected by the ∆*hpf* or the ∆*relA*/∆*spoT* mutations, but the abundance of S13 reduced approximately two-fold from exponential phase to stationary phase (Appendix A).

**Ribosomal proteins are degraded in *P. aeruginosa* ∆*hpf* when cells are starved.** We previously observed a loss of ribosome integrity during starvation of the *P. aeruginosa* ∆*hpf* mutant, by measuring rRNA using fluorescence in situ hybridization (FISH) and by comparing 23S/16S rRNA ratios of starved cells [21]. In those studies, we showed loss of rRNA and a reduction in the 23S/16S rRNA ratio in the ∆*hpf* mutant following starvation, whereas the wild-type strain retained a 23S/16S rRNA ratio of approximately 1.5 throughout starvation. Here, we further characterized ribosome integrity of starved cells by assaying rProtein stability of *P. aeruginosa* and the ∆*hpf* mutant. Ribosomes from *P. aeruginosa* PAO1 and from *P. aeruginosa* ∆*hpf* were partially purified using sucrose cushions at two time-points: (i) stationary-phase cultures (O.D._600_ = 6.0) that were washed and resuspended in PBS and (ii) stationary-phase cultures that were washed and resuspended in PBS, then starved by incubation in PBS for six days. Purified ribosomes were assayed for the rRNA component by agarose gels (Figure 1A), and for the rProtein component by SDS-PAGE (Figure 1B), and by immunoblotting (Figure 1C,D). The rRNA results showed that both the wild-type strain and the ∆*hpf* mutant strain had abundant 16S and 23S rRNAs prior to starvation. Following six days of starvation, both strains had reduced amounts of rRNA, and the 23S and 16S rRNAs were reduced further in the starved ∆*hpf* mutant, compared to the wild-type cells. Almost no rRNA was detected until the sample amount was increased five-fold (Figure 1A, lane 5).

The SDS-PAGE results of purified ribosomes showed similar banding patterns and intensities for the wild-type strain and for the ∆*hpf* mutant prior to starvation (Figure 1B). The protein banding pattern for the wild-type strain was similar in the pre-starved cells and six days-starved cells (Figure 1C, lane 3). In contrast, the banding pattern for the ∆*hpf* mutant strain following starvation differed from the pre-starved cells (Figure 1C, lane 4). When the amount of sample loaded onto the gel for the ∆*hpf* mutant was increased four-fold, the intensity of the protein bands increased in the ∆*hpf* mutant, but the banding pattern of the ∆*hpf* mutant still differed from that of the starved wild-type ribosomes (Figure 1C, lane 5). To characterize specific ribosomal proteins, we performed immunoblot analysis of the partially purified ribosomes, using the L5 and S13 antibodies. The immunoblots showed equivalent amounts of L5 (Figure 1C) and S13 (Figure 1D) in the wild-type strain and the ∆*hpf* mutant strain prior to starvation. Following six days of starvation, there was very little detectable L5 or S13 in ribosomes purified from the ∆*hpf* mutant (Figure 1C,D). Faint L5 and S13 bands could be observed in the ∆*hpf* strain when the amount of protein loaded onto the gel was increased two-fold (Figure 1C,D, lanes 5).

**Ribosomal protein stability in starved *P. aeruginosa* is dependent on HPF.** The experiments using partially purified ribosomes indicated that the amount of both rRNA and rProtein decreased in abundance following *P. aeruginosa* starvation of the ∆*hpf* strain. However, since the ribosome purification approach requires equivalent efficiency in cell lysis and ribosome extraction, the results are only qualitative. In order to quantify ribosomal protein abundances during *P. aeruginosa* starvation, we performed experiments on whole cellular extracts over the course of starvation. These time-course studies allowed sample normalization by loading equivalent amounts of total cellular protein per lane. For these experiments, *P. aeruginosa* PAO1, *P. aeruginosa* ∆*hpf*, *P. aeruginosa* ∆*hpf* + *hpf*, and *P. aeruginosa* ∆*relA*∆*spoT* were cultured to stationary phase (O.D._600_ = 6.0), washed with PBS, then transferred to PBS for starvation at 37 C with shaking for up to eight days. During starvation, cultures were assayed for recoverable cells as colony forming units (CFUs), for total cellular protein, and by immunoblotting for L5 and S13 abundances. All experiments were performed on at least three independent biological replicates, and densitometry was used to compare L5 and S13 band intensities. 

The results of *P. aeruginosa* resuscitation over the course of starvation, quantified as CFUs, were similar to our previous studies [21]. *P. aeruginosa* PAO1 maintained essentially 100% recoverability over eight days of starvation (Figure 2A). In contrast, *P. aeruginosa* ∆*hpf* and *P. aeruginosa* ∆*relA*∆*spoT* were impaired in their ability to recover from starvation, and had approximately ten-fold fewer recoverable cells by 4 d of starvation. Complementation of the *P. aeruginosa* ∆*hpf* with *hpf* on a plasmid restored the ability of the mutant strain to resuscitate from starvation. Interestingly, the results of total cellular protein over the course of starvation differed for the ∆*hpf* mutant versus the ∆*relA*∆*spoT* mutant (Figure 2B). *P. aeruginosa* PAO1 had consistent amounts of cellular protein throughout starvation (*p* = 0.23). Although the ∆*hpf* strain had reduced CFUs over time, the total cellular protein remained similar to that of the wild-type strain (*p* = 0.27). In contrast, the ∆*relA*∆*spoT* mutant lost both recoverable cells (*p* < 0.0001) and total cellular protein (*p* < 0.0001). The results suggest that the ∆*hpf* mutant cells, although impaired in resuscitation, remained intact during starvation, while a majority of the ∆*relA*∆*spoT* cells lysed over the course of starvation, resulting in a decrease in the amount of cellular protein. These results are consistent with our previous microfluidics results that showed that the ∆*hpf* mutant cells remained intact, but not dividing, during starvation [21].

The immunoblotting experiments of cells pre-cultured to early stationary phase (O.D._600_ = 6.0) showed that the wild-type strain, the ∆*hpf* mutant, the ∆*hpf* + *hpf* strain, and the ∆*relA*∆*spoT* mutant strain had similar L5 and S13 levels prior to starvation (Figure 3A). The results of band intensities of triplicate experiments for L5 are shown in Figure 3B and for S13 in Figure 3C. In the wild-type strain, the amounts of L5 and S13 decreased slightly by eight days of starvation (approximately two-fold) (Figure 3A–C). In contrast, the ∆*hpf* strain had approximately ten-fold reduced amounts of L5 and S13 by two days of starvation, and had little detectable L5 and S13 by six days of starvation (Figure 3A–C). These levels were significantly different from the wild type strain (*p* < 0.0001). The amounts of these proteins were restored to wild-type levels in the ∆*hpf* strain complemented with *hpf* on a plasmid (*p* = 0.91 for L5; *p* = 0.20 for S13). The stringent response mutant strain (∆*relA*∆*spoT* strain) had similar levels of L5 (*p* = 0.66) and S13 (*p* = 0.16) to the wild-type strain over the course of starvation (Figure 3).

We next determined if the growth phase prior to starvation affects the subsequent ability of *P. aeruginosa* to resuscitate from starvation. For these experiments, *P. aeruginosa* strains were cultured to exponential phase (O.D._600_ = 1.0) rather than stationary phase, washed in PBS and starved for eight days. Similar to cells precultured to stationary phase, the wild-type strain retained approximately 100% recoverable cells for up to eight days of starvation. The ∆*hpf* mutant and the ∆*relA/*∆*spoT* mutant lost viability faster when precultured to exponential phase than when precultured to stationary phase (Appendix A). Immunoblot experiments using the L5 and S13 antibodies showed that three of the strains, (wild type, ∆*hpf* + *hpf*, and ∆*relA*/∆*spoT*), had reduced abundance of these two proteins after two days of starvation (*p* = 0.0005), then maintained L5 and S13 proteins throughout the remainder of starvation (*p* = 0.33 for L5, *p* = 0.62 for S13)(Figure 3A–C). In the ∆*hpf* mutant, the L5 protein had reduced abundance compared to the wild type, but the cells did not lose L5 to the extent of the ∆*hpf* cells pre-cultured to stationary phase (*p* = 0.0023) (Figure 3B,C). Similarly, S13 had reduced abundance at day 4 of starvation (*p* = 0.04) but was not significantly different from the wild type at the other time points. When cultured to exponential phase, the ∆*hpf* cells did not lose S13 to the extent of cells first cultured to stationary phase (*p* < 0.0001).

Ribosome integrity of the cells pre-cultured to exponential phase (OD_600_ = 1.0) and then starved was assayed as total cellular RNA and as 23S/16S rRNA ratios, prior to starvation and after six days of starvation. The results showed that all strains had reduced cellular RNA content following starvation (*p* < 0.0001) (Figure 4A). Similar to cells cultured to stationary phase [21], the wild-type strain maintained rRNA integrity as assayed by 23S/16S rRNA ratio, maintaining a ratio of approximately 1.4 both before and after six days of starvation (*p* = 0.97) (Figure 4B). The ∆*hpf* strain pre-cultured to exponential phase had reduced rRNA integrity (*p* = 0.0001), with an average 23S/16S rRNA ratio of 0.33 (Figure 4B). The 23S/16S ratio was restored to wild-type levels when the ∆*hpf* strain was complemented with *hpf* on a plasmid (*p* = 0.53). The ∆*relA*∆*spoT* mutation did not have selective loss of the 23S rRNA following starvation (*p* = 0.22). The results indicate that when *P. aeruginosa* is pre-cultured to exponential phase, then starved, HPF is required to maintain rRNA integrity. In addition, the abundances of rProteins is reduced in the ∆*hpf* strain during starvation, but not to the extent of rProtein loss in cells first cultured to stationary phase, then starved. 

**Nutrient sources affect *hpf* expression and subsequent resuscitation of *P. aeruginosa* from starvation.** Previously, we identified the *hpf* promoter sequence and characterized factors involved in *hpf* expression [41]. For those studies, we generated a yellow fluorescent protein (YFP) reporter gene to assay for expression from the P*_hpf_* promoter (P*_hfp_*-*hpf*-*yfp*) [41]. *P. aeruginosa* PAO1 grows in minimal medium with a variety of simple sugars as sole carbon sources, but with varying growth rates depending on the sugar. Here, we used the P*_hfp_*-*hpf*-*yfp* reporter to compare the expression of *hpf* in cells cultured in defined minimal medium (MOPS medium) with either glucose or fructose as the sole carbon source. In MOPS-fructose medium, *P. aeruginosa* P*_hfp_*-*hpf*-*yfp* had an extended lag time compared to growth in MOPS-glucose medium, and in MOPS-fructose, expression from the P*_hpf_* promoter closely followed cell growth (Figure 5). In contrast, in MOPS-glucose medium, the lag time was short, and there was an additional lag between growth and expression from the P*_hpf_* promoter (Figure 5). Therefore, culturing *P. aeruginosa* in minimal medium with either glucose or fructose as the sole carbon should result in differing cellular levels of HPF, and possibly differing abilities of the cells to recover from starvation. To test this, we cultured strains in MOPS-fructose and MOPS-glucose to early exponential phase (O.D._600_ = 0.3 or O.D._600_ = 1.0). Following growth, the cultures were starved as described above, by washing in PBS, resuspending equivalent cell numbers in PBS (approximately 2 x 10^8^ cells/mL), and incubating in PBS at 37 C for up to eight days. The starved cultures were assayed for resuscitation from starvation, total cellular protein, 23S/16S rRNA ratios, and relative amounts of HPF, L5, and S13. 

In order to compare strains and growth conditions, the reduction in CFUs was plotted over time of starvation, and the results were compared to those from cells pre-cultured in rich medium (TSB) to O.D._600_ = 1.0 (Figure 6). Under all pre-growth conditions, *P. aeruginosa* PAO1 increased in cell numbers after one day of starvation (Figure 6). This phenomenon was observed previously, and is due to reductive cell division early in starvation [21]. For the wild-type strain, cells cultured in MOPS-glucose had an approximately 2.5 fold reduction in cell recovery by day eight of starvation (*p* < 0.0001), whereas there was no reduction in CFU when cells were first cultured in rich medium (*p* = 0.63) and little reduction in recovery in *P. aeruginosa* PAO1 cultured in MOPS-fructose (Figure 6A). Under all four pre-growth conditions, the ∆*hpf* mutant lost recoverable cells over the course of starvation (*p* < 0.0001) (Figure 6B). Complementation of the *P. aeruginosa* ∆*hpf* mutant with *hpf* under control of a strong promoter restored resuscitation to levels similar to the wild-type strain (Figure 6C). 

The assay for cellular protein over the course of starvation showed reduction in total cellular protein by two-fold over eight days of starvation regardless of the pre-growth condition or the strain (Appendix A). The results of immunoblots show that HPF is produced early in cell growth (day 0, prior to starvation). Surprisingly, even when cells are cultured in MOPS-glucose to early exponential phase (O.D._600_ = 0.3), a band associated with HPF is detected (Figure 7A,B). No HPF band is observed in the ∆*hpf* strain, and an intense band is observed in the ∆*hpf* strain where *hpf* is expressed from a plasmid and strong promoter. Under all four pre-growth conditions, HPF is maintained in cells through eight days of starvation (*p* = 0.57) (Figure 7A–C). *P. aeruginosa* ∆*hpf* with *hpf* expressed from a strong promoter had high levels of HPF initially, but the abundance of HPF reduced to wild-type levels by two days of starvation and then remained at wild-type levels throughout starvation, suggesting that the excess HPF is degraded during *P. aeruginosa* starvation. 

We next determined if the pre-cultivation conditions prior to starvation affected rProtein stability of the large subunit or small subunit ribosomal proteins (Figure 8 and Figure 9). In the wild-type strain, the cellular L5 protein abundance reduced slightly over the course of starvation (Figure 8). In contrast, L5 showed significantly reduced abundance over the course of starvation in the ∆*hpf* strain regardless of the pre-cultivation conditions (*p* < 0.0001, MOPS-glucose, MOPS-fructose, O.D._600_ = 0.3 and 1.0) (Figure 8). Complementation of *hpf* in the ∆*hpf* strain restored the L5 levels over the course of starvation to wild-type levels. Similarly, S13 was maintained in the wild-type strain and in the complemented strain throughout starvation (Figure 9). The ∆*hpf* strain had reduced abundance of S13 over the course of starvation (*p* = 0.0002), regardless of the pre-cultivation conditions (Figure 9A–C). However, both L5 and S13 were detected even after eight days of starvation, when the cells were first cultured in minimal medium prior to starvation.

We determined if pre-growth of strains to early exponential phase in MOPS-glucose or MOPS-fructose influenced rRNA integrity following starvation, by measuring total cellular RNA (Figure 10A) and the 23S/16S rRNA ratios before and after six days of starvation (Figure 10B). Prior to starvation, all strains had similar 23S/16S rRNA ratios regardless of the pre-growth conditions (*p* = 0.90) (Appendix A). Following six days of starvation, the wild-type strain had preserved 23S/16S rRNA ratios of approximately 1.5 (Figure 10B). However, the ∆*hpf* strain had selective loss of the 23S rRNA component in the ∆*hpf* strain compared to the wild-type strain after six days of starvation (*p* < 0.0001). The results show that, even when *P. aeruginosa* is cultured to early exponential phase in minimal medium where *hpf* expression is low, there is sufficient HPF to protect against subsequent loss of rRNA and rProteins during starvation.

## 3. Disscussion

Heterotrophic bacteria are subject to changing environmental conditions, including periods where nutrients are abundant, as well as periods of starvation. Biofilms contain nutrient gradients, due to diffusion of nutrients into the biofilms and utilization of the nutrients by the bacteria [42]. As a result, biofilms contain bacteria in many different physiological states [6], including cells that are nutrient-limited. The biofilm-forming bacterium, *P. aeruginosa*, has adapted to tolerate nutrient-deficient conditions, and can survive starvation for long periods of time with little loss of recoverable cells [21]. Like most heterotrophic bacteria, *P. aeruginosa* responds to its changing environmental conditions, including shifts from nutrient abundance to nutrient starvation, through a variety of metabolic processes [13,14]. Among these changes are conservation of energy resources and degradation of excess cellular proteins and RNA. Ribosomes are an excellent source of resources, including RNA and proteins, that can be used as nitrogen, phosphate, and energy sources under nutrient limiting conditions. As a result, when cells transition from rapid growth to slow growth, they reduce their number of ribosomes [43], presumably to a maintenance level necessary for resuscitation when nutrients become available [15]. In a previous study, we characterized rRNA stability and abundance during *P. aeruginosa* starvation [21]. In the study here, we further characterized the fate of ribosomes during *P. aeruginosa* starvation, by assaying protection or loss of two essential ribosomal proteins, L5 and S13, following the transition of *P. aeruginosa* from active growth to starvation. The results show that these proteins are stably maintained in *P. aeruginosa* during starvation, and that HPF is required for the maintenance of these proteins. The results also show that, although the stringent response is essential for optimal recovery of *P. aeruginosa* from starvation, the stringent response is not required for rProtein maintenance during starvation. Therefore, the mechanisms by which HPF and the stringent response protect *P. aeruginosa* viability during starvation differ.

The pathway for ribosome degradation has been well characterized for *E. coli* [34,43,44,45]. Ribosome degradation is initiated by RNAse E endonuclease cleavage at positions 820 and 919 of the 16S rRNA and position 1942 of the 23S rRNA [34]. The rRNA fragments are then further processed by other ribonucleases, including PNPase, RNAse R, RNAase II, and RNAse R. Cleavage and degradation of rRNA causes the release of ribosomal proteins, which may then either be targeted for degradation by proteases, including Lon protease [46], or recycled for incorporation into new ribosomes [47]. The Lon protease is activated by polyphosphate [46], which accumulates as a starvation response in *P. aeruginosa* [48]. HPF likely evolved to prevent the initial events in ribosome degradation [34,35,44], preserving a subset of ribosomes in the inactive 70S or 100S form. Here, we show that protection of rRNA and rProtein degradation is dependent on the presence of HPF. The loss of rRNA is independent of the growth conditions prior to starvation. Regardless of the pre-growth cultivation conditions, in *P. aeruginosa* ∆*hpf* rRNA loses integrity during subsequent starvation, as measured by total RNA and by the reduced 23S/16S rRNA ratio (Figure 4 and Figure 10). The loss of rRNA integrity correlates with the impaired ability of *P. aeruginosa* to recover from starvation. In contrast, the degree of rProtein loss in *P. aeruginosa* ∆*hpf* is dependent on the physiology of the cells prior to entering starvation. The greatest loss of L5 and S13 in the *P. aeruginosa* ∆*hpf* mutant occurs when the cells are pre-cultured to stationary phase in rich medium. Polyphosphate is expected to accumulate in cells cultured to stationary phase, resulting in activation of the Lon protease [46], and greater rProtein degradation. Interestingly though, when HPF is overexpressed prior to starvation in cells pre-cultured in balanced minimal medium, the excess HPF is reduced to wild-type levels during the subsequent cell starvation. This suggests that even when the cells are not in stationary phase prior to starvation, excess HPF (possibly the HPF not associated with ribosomes) is targeted for degradation when the cells enter starvation.

In *P. aeruginosa*, and in many other bacteria, the *hpf* gene is linked to the *rpoN* gene [41]. *rpoN* encodes an alternative sigma factor, σ^N^, which is a master regulator for the nitrogen-starvation response. However, *hpf* also contains its own promoter (P*_hpf_*) that is located within the *rpoN* coding sequence [41]. The P*_hpf_* promoter may allow cells to express high levels of *hpf* even when nitrogen is not limiting. Here, we tested the expression of *hpf* when cells were cultured to early exponential phase in balanced minimal medium, where the cells should not be limited for nitrogen, phosphate, or carbon. The results show that even in early exponential phase in nitrogen-replete conditions, *hpf* is expressed and detectable levels of HPF protein are produced (Figure 5 and Figure 7). The HPF that is produced in early exponential phase is necessary for optimal recovery of *P. aeruginosa* from subsequent starvation and for protection of ribosome integrity during starvation. Therefore, a subset of ribosomes contain HPF even in rapidly growing bacteria. Ribosome degradation is not restricted to stationary-phase cells or to cells with excess ribosomes. Ribosome degradation is also associated with quality control [44]. Ribosome assembly is a complex process, and ribosomes must pass certain quality control checkpoints prior to becoming translationally active molecules. Ribosomes that do not pass quality control are either degraded or the damaged ribosomal proteins are replaced [47]. Since *hpf* is expressed throughout growth and even in nitrogen-replete medium, it is possible that HPF not only plays a role in protecting ribosomes during cell starvation, but also in the ribosome assembly and quality control process.

In summary, the results here show that HPF and the stringent response are required for optimal recovery of *P. aeruginosa* from starvation. The stringent response plays a role in preventing cell lysis when the cells enter starvation. HPF is required to prevent the early steps in ribosome degradation, including loss of rRNA and loss of rProteins, which lead to insufficient ribosomes necessary for cell recovery from starvation.

## 4. Materials and Methods

**Strains.***P. aeruginosa* PAO1 and its derivatives were used in these studies. Mutant strains included *P. aeruginosa* PAO1 ∆*hpf* [21], *P. aeruginosa* PAO1 ∆*hpf* + *hpf*, which carries the wild type copy of *hpf* on the IPTG-inducible vector pMF54 [21,49], and *P. aeruginosa* Δ*relA*Δ*spoT* [50]. For overexpression of the 6X-poly-histidine fusion proteins from plasmid pET28a, *E. coli* BL21 (DE3) was used. PAO1::P*_hpf_*-HPF-YPF, PAO1::P*_BAD_*-L5-YFP, PAO1::P*_BAD_*-L5-6xHis, and PAO1::P*_BAD_*-S13-YFP were used for antibody verification experiments. PAO1::P*_hpf_*-HPF-YPF was also used in growth and *hpf* gene expression experiments [41].

**Media and buffers.** Tryptic Soy Broth and Tryptic Soy Agar (Difco, Sparks, MD, USA) were used for general cultivation of *P. aeruginosa* and *E. coli*. Isopropyl β-d-1-thiogalactopyranoside (IPTG) (1 mM), carbenicillin (150 μg/mL) and kanamycin (30 μg/mL) were added to media when necessary. MOPS buffer contained: 40 mM 3-N-morpholino-propanesulfonic acid (MOPS), 4.0 mM tricine, 0.01 mM FeSO_4_, 0.276 mM K_2_SO_4_, 0.5 µM CaCl_2_, 0.02 mM MgCl_2_, 50 mM NaCl, 132 mM K_2_PO_4_, pH 7.2. MOPS minimal medium contained MOPS buffer plus 20 mM NH_4_Cl and 100 μL/L micronutrients, supplemented with either 40 mM glucose or 40 mM fructose. The micronutrient solution contained 0.146 mM ammonium molybdate, 20.05 mM boric acid, 1.51 mM cobalt chloride, 0.481 mM cupric sulfate, 0.404 mM manganese chloride, and 0.487 mM zinc sulfate. Phosphate buffered saline (PBS) contained 0.127 M NaCl, 7.0 mM Na_2_HPO_4_, and 3 mM NaH_2_PO_4_ (pH 7.0).

**Plasmid constructs.** For purification of ribosomal proteins, the genes for *rplE* and *rpsM* (encoding ribosomal proteins L5 and L13) were cloned into the poly-histidine expression vector pET28a to produce C-terminal 6x-poly-histidine tags, and *hpf* was purified using the 6x-poly-histidine tag as described previously [22,51]. Restriction endonucleases and DNA ligase were purchased from New England Biolabs (Ipswich, MA, USA). PCR was used to amplify the three genes from *P. aeruginosa* genomic DNA using the primers: *hpf*-forward CGC ATA TGC AAG TCA ACA TCA GTG GCC ATC, *hpf*-reverse AGC TCG AGT CAG CGG GCG CCT ACG CCT TGC5; L5-forward GGC ATA TGG CAC GAT TGA AAG AAA TTT ATC and L5-reverse CGC TCG AGC TCG CGG TTC TTC ATG CTC TC; S13-forward GGC ATA TGG CCC GTA TTG CAG GCG TCA AC and S13-reverse GGC TCG AGC AGC AGG TTT TGC CAT GAC TAG. PCR products were ligated in-frame with the 6x-poly-histidine tags using the *Nde*I and *Xho*I restriction sites.

**Protein expression and purification.** 6x-poly-histidine tagged L5 and S13 were expressed and purified from *E. coli* BL21 (DE3) cells containing pET28a-based plasmids. LB medium (Difco, Sparks, MD, USA) (1 L) supplemented with 30 μg/mL kanamycin was inoculated with 10 mL of overnight cultures, then incubated at 37 °C with shaking at 200 rpm for 8 h. After 8 h, 1 mM IPTG (Isopropyl β-D-1-thiogalactopyranoside) was added, and the cultures were incubated for an additional 16 h. Cell pellets were harvested by centrifugation at 3000× *g* for 15 min at 4 °C, then frozen at −20 °C. Cell pellets were resuspended in 10 mL of lysis buffer (100 mM Tris-HCl pH ~ 8, 150 mM sodium chloride, 5 mM imidazole) and disrupted by sonication with a Branson Digital Sonifier (Branson Ultrasonics, Brookfield, CT, USA) on ice. Cell lysates were then centrifuged at 10,000× *g* at 4 °C for 30 min to remove cellular debris. The supernatants containing the soluble proteins were collected and purified by cobalt affinity chromatography on HisPur^TM^ Cobalt Resin (Thermo Fisher Scientific, Rockford, IL, USA) according to the manufacturer’s instructions for native conditions. 6x-poly-histidine tagged proteins were eluted with Elution Buffer containing 50 mM sodium phosphate, 300 mM sodium chloride, and 300 mM imidazole at pH 7.4 in ten 2 mL fractions. The purified protein eluates were monitored by the Bicinchoninic Acid (BCA) Assay (Pierce Biotechnology, Waltham, MA, USA), absorbance at 562 nm on the Nanodrop1000 (Thermo Fisher Scientific, Rockford, IL, USA), and SDS-PAGE using Coomassie blue staining. Additionally, proteins were assayed with immunoblots using an antibody to the 6x-poly-histidine tag (Thermo Fisher Scientific, Rockford, IL, USA) with chemiluminescent detection, as described previously [52].

**Antibody Generation and Purification.** Poly-clonal antisera to L5 and S13 proteins were raised in rabbits by Lampire Biological Laboratories (Pipersville, PA, USA). The IgG fraction of the antisera was purified using NAb^TM^ Protein A Plus Spin Columns (Thermo Fisher Scientific, Rockford, IL, USA) according to the manufacturer’s instructions. Polyclonal antisera to HPF was purified as described previously [22]. The efficacies of the purified antibodies were tested using immunoblots against the purified protein, against *P. aeruginosa* PAO1 protein extracts, and against extracts of *P. aeruginosa* containing fusions of the ribosomal proteins to the yellow fluorescent protein.

**Ribosome purification and analysis.** Ribosomes were partially purified from *P. aeruginosa* PAO1 and *P. aeruginosa* ∆*hpf* cells before and after starvation in phosphate buffered saline (PBS) pH 7.0. For these experiments, 1 mL of overnight culture was added to 100 mL Tryptic Soy Broth (TSB). The cultures were incubated at 37 °C with 200 rpm shaking until the OD_600_ reached 6.0 on a CE2041 Spectrophotometer (Cecil Instrumentation Services Ltd., Milton, Cambridge, UK). Cells were washed twice in PBS, resuspended in 10 mL PBS, and then added to 100 mL of PBS to give a final OD_600_ of 1.0, corresponding to approximately 2.4 × 10^9^ CFU/mL. The 110 mL samples were collected immediately or following 6 days of incubation with shaking at 200 rpm at 37 °C. For sample collection, the cells in PBS were centrifuged at 8000× *g* for 10 min at 4 °C, washed once in PBS, and transferred to 1.5 mL microcentrifuge tubes. Cells were further concentrated by centrifugation at 9500× *g* for 5 min at 4 °C. The supernatant was removed and the cell pellets were frozen at −80 °C. The cells were resuspended in 50 µL ice-cold Buffer A [53], brought to 1 mL with a final concentration of 0.7 mM lysozyme, 1 mM phenylmethylsulfonyl fluoride (PMSF), 0.5× CelLytic^TM^ B (Sigma-aldrich, Burlington, MA, USA), 3 mM Magnesium, and diethyl pyrocarbonate (DEPC) treated water. Cells were lysed by eight freeze-thaw cycles consisting of 10 min at −80 °C followed by 2 min in a 37 °C water bath. Ten units of RNase-Free DNAse (New England Biolabs, Ipswich, MA, USA) was added to each cell lysate and five additional freeze-thaw cycles were performed to ensure complete lysis. The cell lysates were spun at 20,000× *g* for 45 min at 4 °C to remove cellular debris. The supernatants were collected and ribosomes were separated using Buffers A, B, and C from [53] according to the previously published crude preparation protocol in [54]. Briefly, the supernatants were adjusted to a final volume of 6 mL with Buffer A and slowly added to RNase-free ultracentrifuge tubes (Beckman Coulter, Indianapolis, IN, USA) containing 6 mL Buffer B. The fractions were centrifuged at 111,132× *g* for 15 h at 4 °C using the SW41 Ti rotor in an Optima^TM^ L-100 XP Ultracentrifuge (Beckman Coulter, Indianapolis, IN, USA). The supernatants were carefully removed, and the ribosome pellets were resuspended in 200 µL Buffer C. 

To assay for rRNA in ribosome preparations, 2 μL aliquots of purified ribosomes were run on agarose gels containing bleach as described previously [55]. To assay the ribosomal proteins of purified ribosomes, aliquots (10 µL) of purified ribosomes were separated by SDS-polyacrylamide gels (16%) and stained with Coomassie Brilliant Blue R-250 [52]. For immunoblotting of partially purified ribosomes, 10 µL aliquots of the purified ribosome were separated on SDS-PAGE then wet transferred to nitrocellulose membranes (GE Healthcare Life Sciences, Marlborough, MA, USA) at 350 mA for one h. The membranes were washed, blocked, and incubated with polyclonal primary antibodies in 3% skim milk in TBST (Tris buffered saline, 0.1% Tween) overnight on a rocking platform. The membranes were then washed with TBST four times for 15 min each, then incubated in polyclonal rabbit conjugated Horseradish Peroxidase secondary antibody (Thermo Fisher Scientific, Rockford, IL, USA) in 3% Bovine Serum Albumin in TBST for at least three h. Membranes were then washed and developed. Chemiluminescent detection of x-ray autoradiography film (Santa Cruz Biotechnology, Dallas, TX, USA) was performed with an SRX-101A Medical Film Processor (Konica Minolta Medical & Graphic, Inc., Wayne, MJ, USA). Each immunoblot experiment was performed on at least three independent biological replicates.

**Ribosomal protein analysis during *P. aeruginosa* growth.** Overnight cultures of *P. aeruginosa* PAO1, *P. aeruginosa* ∆*hpf*, *P. aeruginosa* ∆*hpf* + *hpf*, and *P. aeruginosa* ∆*relA*∆*spoT* were used to inoculate 25 mL of TSB to achieve an OD_600_ of 0.01. Carbenicillin and IPTG were added to the medium for the *P. aeruginosa* ∆*hpf* + *hpf* strain. The cultures were incubated at 37 °C with shaking at 200 rpm. Samples (1 mL) were collected after 0, 2, 4, 6, 8, 10, 12, and 24 h of incubation by centrifugation 7690× *g* for 3 min at 4 °C. Cell pellets were lysed by incubation in 100 μL Radio Immunoprecipitation Assay Lysis buffer (RIPA) pH 7.6, containing 10 mM Tris-Base, 1 mM EDTA, 140 mM sodium chloride, 0.1% Sodium deoxycholate (DOC), 0.1% sodium dodecyl sulfate (SDS), and 1% NP-40, at 85 °C for five min. To degrade nucleic acid, 1 μL of Pierce Universal Nuclease (Pierce Biotechnology, Waltham, MA, USA) was added and samples were incubated at room temperature for 5 min. Cellular protein amounts were then determined using the BCA assay (Pierce Biotechnology, Waltham, MA, USA ). The mean and standard deviation of total cellular protein recovered from three biological replicates were plotted using GraphPad Prism v 8.3 (GraphPad, San Diego, CA, USA). For samples collected between 4 and 24 h, 15 μg of total protein from each sample was separated on 16% SDS-PAGE gels, wet transferred to nitrocellulose membranes, and probed with the antibodies generated against HPF, S13, and L5. Developed x-ray films were imaged and band intensities were quantified using Image Studio Software (LI-COR Biosciences, Lincoln, NE, USA). GraphPad Prism was used to plot the mean and standard error of band intensities from three biological replicates. For these assays, GraphPad Prism was also used to perform 2-way ANOVA with Dunnett’s test to correct for multiple testing, when necessary. For all statistical tests, an α of 0.01 was used, and multiplicity adjusted *p* values were reported.

**Nutrient starvation assays.** Starved cultures were prepared as described previously [21]. For all strains and media conditions the starved cultures were prepared to achieve approximately 2 × 10^8^ cells/mL as the initial cell concentration in the starvation cultures. For starvation following growth in TSB, overnight cultures of *P. aeruginosa* and its mutant derivatives were used to inoculate flasks containing 60 mL TSB. For the *hpf* complemented strain, overnight cultures were supplemented with 150 µg/mL carbenicillin and 1 mM IPTG. The 60 mL cultures were incubated at 37 °C with aeration until the OD_600_ reached 1.0 (exponential phase) or 6.0 (stationary phase). Aliquots of cells containing approximately 10^10^ CFU were removed, washed twice with PBS pH 7.2, resuspended in PBS, and added to 60 mL of PBS to achieve 2 × 10^8^ cells/mL. The resulting starvation cultures were incubated at 37 °C with shaking at 200 rpm for eight days, and sampled daily for use in cell viability, protein, and RNA assays. Starvation cultures following growth in MOPS-minimal media were prepared similarly, with the following modifications. Overnight cultures were washed twice in MOPS buffer then used to inoculate flasks containing 100 mL MOPS minimal media supplemented with either 40 mM fructose or 40 mM glucose, to O.D._600_ = 0.05. Cells were incubated in 100 mL MOPS-fructose or MOPS-glucose until the OD_600_ reached 0.3 or 1.0. At this point, approximately 10^10^ CFU were removed by centrifugation, washed twice with PBS pH 7.2, resuspended in PBS to approximately 2 × 10^8^ CFU/mL. Starvation cultures were incubated with shaking at 200 rpm at 37 °C and sampled daily for use in cell viability, protein, and RNA assays.

**Determination of cell viability.** Over the course of growth and starvation, the drop plate method was used to quantify colony forming units (CFUs). Briefly, cells were serially diluted in 0.85% NaCl and replicate 10 µl volumes were plated on TSA. Colonies were counted after 18 h incubation at 37 °C. For fold-reduction over time, CFUs were normalized by dividing by the number of cells at zero days of starvation. The mean and standard deviations from at least three biological replicates per experiment were plotted in GraphPad Prism v8.3. Two-way ANOVA with Dunnett’s correction for multiple comparisons was performed to determine statistically significant differences at α = 0.01.

**Protein analysis during starvation conditions.** On days 0, 2, 4, 6, and 8 of starvation, volumes ranging from 4.5 to 9 mL were removed from starvation cultures. The cells were collected by centrifugation at 7690× *g* for three min at 4 °C, the supernatants were removed, and cell pellets were immediately frozen at −80 °C. Cells were then lysed in 100 μL RIPA buffer and total protein was quantified as described previously for growth studies. For comparison of strains cultured in different media, the reduction of cellular protein over starvation time was plotted as the change from the initial time point. For immunoblot analysis, 15 µg of total cellular protein from each sample at each timepoint was loaded per lane for SDS-PAGE analysis. Immunoblotting and densitometry of the ribosomal protein bands was performed as described above. The mean and standard error from at least three independent biological replicates were plotted in GraphPad Prism v 8.3. 2-way ANOVA was performed with Dunnett’s or Sidak’s multiple comparison test, as appropriate. Multiplicity adjusted *p*-values were reported at α of 0.01.

***P. aeruginosa hpf* expression from P*_hpf_*.** To assay for *hpf* expression over the course of growth, overnight cultures of *P. aeruginosa* PAO1::P*_hpf_*-HPF-YPF were centrifuged at 9500× *g* for one min. The cell pellets were washed twice with MOPS buffer pH 7.2, then used to inoculate flasks containing 50 mL MOPS minimal media supplemented with either 40 mM fructose or 40 mM glucose to an initial OD_600_ of 0.05. Cultures were incubated at 37 °C with shaking at 200 rpm. Samples were collected over 48 h to assay for growth as optical density at 600 nm on the Ultrospec 2100 Pro (Amersham Biosciences, Little Chalfont, UK) and for HPF production as YFP fluorescence on the Cytation 5 plate reader (BioTek Instruments, Winooski, VT, USA). This experiment was performed on three independent biological replicates.

**RNA extraction and analysis.** During starvation, cell aliquots (0.6 mL on day zero and 1.8 mL on day six) were collected by centrifugation at 7700× *g* for 2 min at 4 °C. The supernatants were quickly removed, and cell pellets were frozen at −80 °C until RNA was extracted, which proceeded as described in [21]. Briefly, the hot phenol method, with an eight-minute 65 °C incubation, was used to extract RNA. Following a second phase separation step utilizing a heavy phase lock gel (5 Prime, QuataBio, Beverly, MA, USA), the Clean and Concentrator-5 kit (Zymo Research, Irvine, CA, USA) was used according to the manufacturer’s instructions. RNA yield was measured in triplicate for each sample on the NanoDrop1000 (Thermo Fisher Scientific, Rockford, IL, USA) on a minimum of three biological replicates. The mean and standard error of the nanograms of RNA recovered per µL of starvation culture were plotted using GraphPad Prism 8.3.0. To assess integrity, RNA samples were then visualized on the Bioanalyzer 2100 (Agilent Technologies, Santa Clara, CA, USA) using the Prokaryotic Total RNA 6000 Nanoassay (Agilent Technologies). The mean and standard error of the ratio of 23S rRNA to 16S rRNA were plotted using GraphPad Prism 8.3.0 for a minimum of three biological replicates. GraphPad Prism was also used to perform 2-way ANOVA with Sidak’s multiple comparison test at *alpha* = 0.01 to determine statistically significant differences.

## Figures and Tables

**Figure 1 ijms-21-09494-f001:**
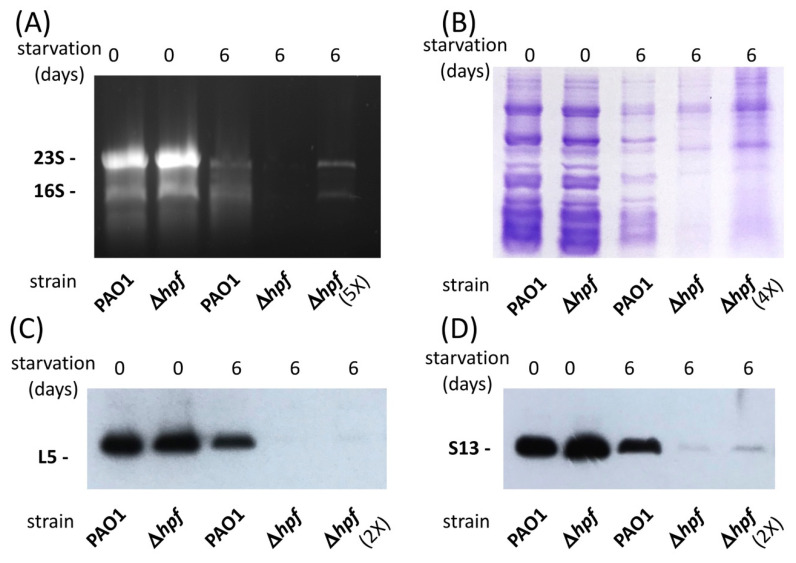
Analysis of *P. aeruginosa* ribosomes, partially purified using sucrose cushions. Ribosomes were purified from pre-starved cells and from 6 d starved cultures. (**A**) rRNA extracts of purified ribosomes from *P. aeruginosa* and from the ∆*hpf* strain prior to starvation and after 6 d of starvation. (**B**) SDS-PAGE of purified ribosomes of pre-starved and 6 d starved cultures. (**C**) Immunoblot analysis of protein L5 from pre-starved cultures and 6 d starved cultures. (**D**) Immunoblot analysis of protein S13 from pre-starved cultures and from 6 d starved cultures. Shown are representative images from three independent replicate gels.

**Figure 2 ijms-21-09494-f002:**
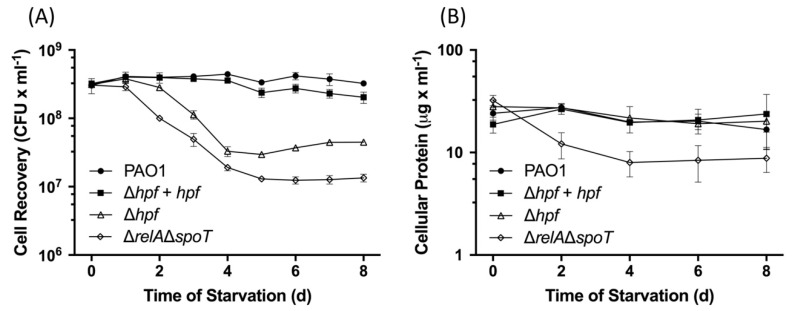
(**A**) Resuscitation of *P. aeruginosa* PAO1, *P. aeruginsoa* ∆*hpf*, *P. aeruginsoa* ∆*hpf* complemented with *hpf* on a plasmid, and *P. aeruginsoa* ∆*relA*∆*spoT* over the course of starvation at 37 C in phosphate buffered saline (PBS). Cultures were diluted to give approximately the same initial concentration of cells. Resuscitation was assayed as colony forming units (CFUs). (**B**) Cellular protein abundances of *P. aeruginosa* strains over the course of starvation. Data show the mean and standard deviation for three replicates.

**Figure 3 ijms-21-09494-f003:**
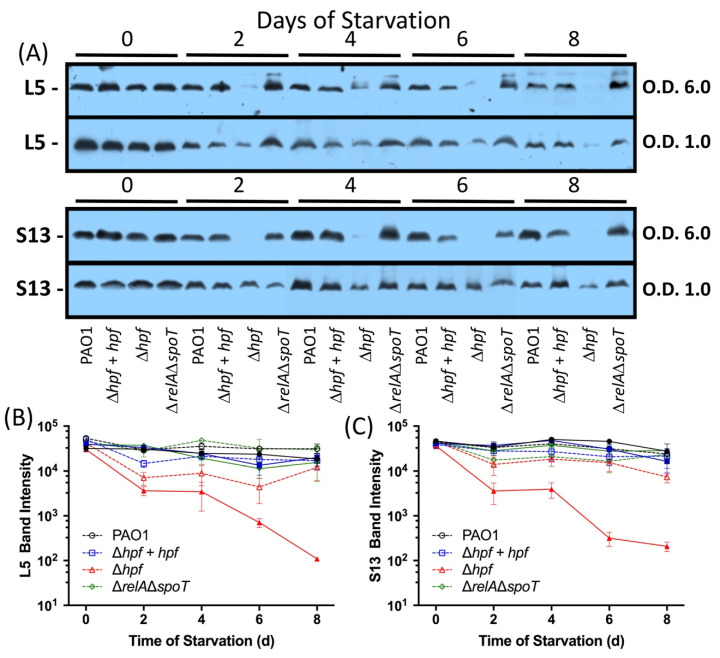
Immunoblot and densitometry analysis of *P. aeruginosa* L5 and S13 ribosomal proteins over the course of starvation. (**A**) L5 and S13 Immunoblots of *P. aeruginosa* PAO1, *P. aeruginosa* ∆*hpf*, *P. aeruginosa* ∆*hpf* + *hpf* and *P. aeruginosa* ∆*relA*∆*spoT* over 8 d of starvation. Strains were pre-cultured to O.D._600_ = 6.0 (filled symbols and solid lines) or O.D._600_ = 1.0 (open symbols and dashed lines) prior to starvation. (**B**) Densitometry of L5 immunoblots showing the mean and standard error of the mean of three replicate immunoblots. (**C**) Densitometry of S13 immunoblots showing the mean and standard error of the mean of three replicate immunoblots.

**Figure 4 ijms-21-09494-f004:**
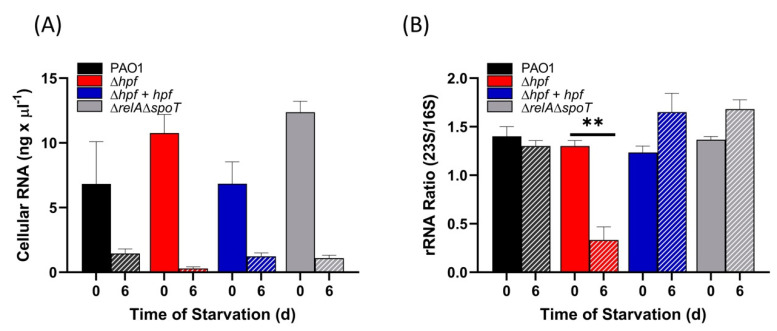
rRNA analysis of *P. aeruginosa* and mutant strains prior to starvation (solid bars) and after 6 d of starvation (striped bars). (**A**) Total RNA abundances of pre-starved and 6 d starved cells assayed using the NanoDrop Spectrophotometer. (**B**) 23S/16S rRNA ratios of cells prior to starvation and following 6 d starvation, determined using the Agilent Bioanalyzer. Data show the mean and standard error of the mean for at least three independent replicates. Asterisks (**) for 23S/16S ratio shows significant difference between 0 and 6 d (*p* < 0.001).

**Figure 5 ijms-21-09494-f005:**
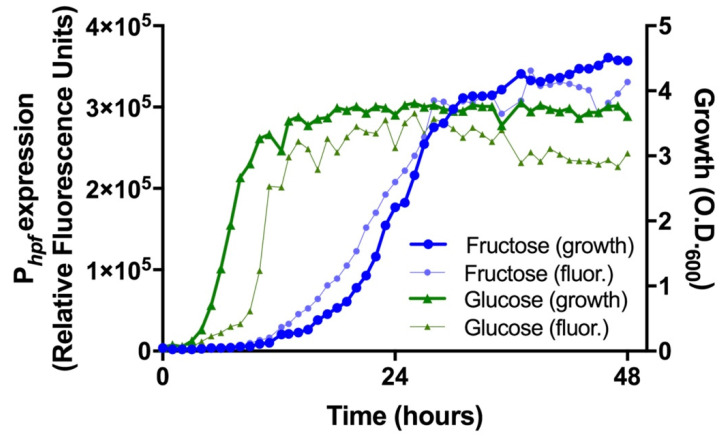
Reporter expression of P*_hpf_* when *P. aeruginosa* is cultured in MOPS minimal medium with glucose or fructose as the sole carbon source. *P. aeruginosa* was assayed for growth as as O.D._600_ and for *hpf* expression as fluorescence from the P*_hpf_*-*hpf*-*yfp* single-copy reporter. Green symbols show growth and fluorescence in glucose and blue symbols show growth and fluorescence in fructose.

**Figure 6 ijms-21-09494-f006:**
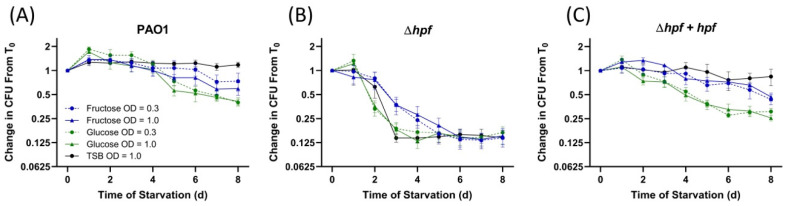
Effect of nutrient source and pre-growth conditions on subsequent resuscitation of *P. aeruginosa* from starvation. Strains were cultured in MOPS minimal medium with glucose (green) or fructose (blue) to O.D._600_ = 0.3 (dashed lines) or O.D._600_ = 1.0 (solid lines). Data are plotted as fold-reduction over 8 d of starvation and compared to strains pre-cultured in TSB to O.D._600_ = 1.0 (black lines). (**A**) Fold-change in recoverable cell numbers for the wild-type strain, *P. aeruginosa* PAO1. (**B**) Fold-change in recoverable cells for *P. aeruginosa* ∆*hpf*. (**C**) Fold-change in recoverable cells for *P. aeruginosa* ∆*hpf* complemented with *hpf*. All data show the mean and standard error of at least three independent biological replicates.

**Figure 7 ijms-21-09494-f007:**
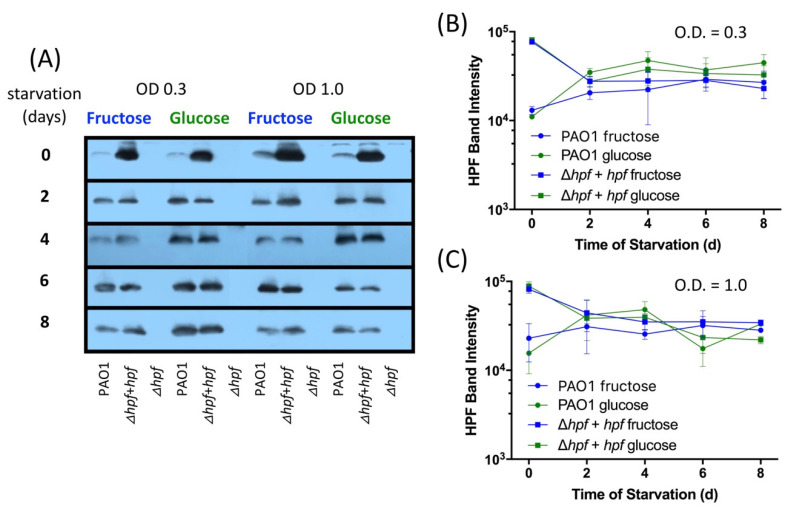
Effect of nutrient source and pre-growth conditions on cellular HPF abundances. (**A**) Immunoblots of HPF over 8 d of starvation for strains *P. aeruginosa* PAO1, ∆*hpf*, ∆*hpf* + *hpf*, pre-cultured in minimal medium with glucose or fructose to early exponential phase O.D._600_ = 0.3 or O.D._600_ = 1.0. (**B**) Mean and standard error of the mean for immunoblot HPF band intensities for PAO1 and ∆*hpf* + *hpf* strains pre-cultured in minimal medium to O.D._600_ = 0.3. (Results from ∆*hpf* strain were below detection). (**C**) Mean and standard error of the mean for immunoblot HPF band intensities for PAO1 and ∆*hpf* + *hpf* strains pre-cultured in minimal medium to O.D._600_ = 1.0.

**Figure 8 ijms-21-09494-f008:**
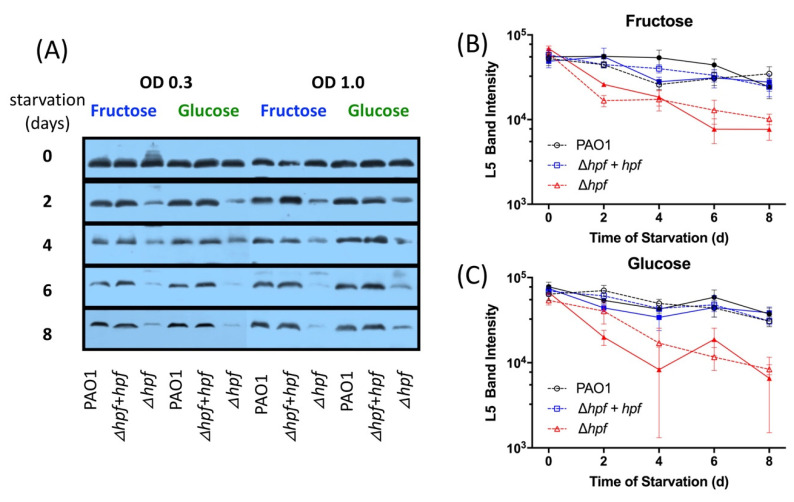
Quantification of *P. aeruginosa* L5 band intensity over the course of 8 d of starvation, from whole cell extracts. Strains were pre-cultured in MOPS minimal media to early exponential phase (O.D._600_ = 0.3 or 1.0) with either fructose or glucose as the sole carbon source. (**A**) Representative immunoblot of L5 from whole cell extracts. (**B**) Quantification of L5 signal intensity obtained by densitometry of three immunoblots, when strains were cultured in MOPS-fructose. Solid lines and symbols indicate cells were pre-cultured to O.D._600_ = 1.0 and dashed lines and open symbols indicate cells were pre-cultured to O.D._600_ = 0.3. (**C**) Quantification of L5 signal intensity when strains were cultured in MOPS- glucose. Data show the mean and standard error of three independent biological replicates.

**Figure 9 ijms-21-09494-f009:**
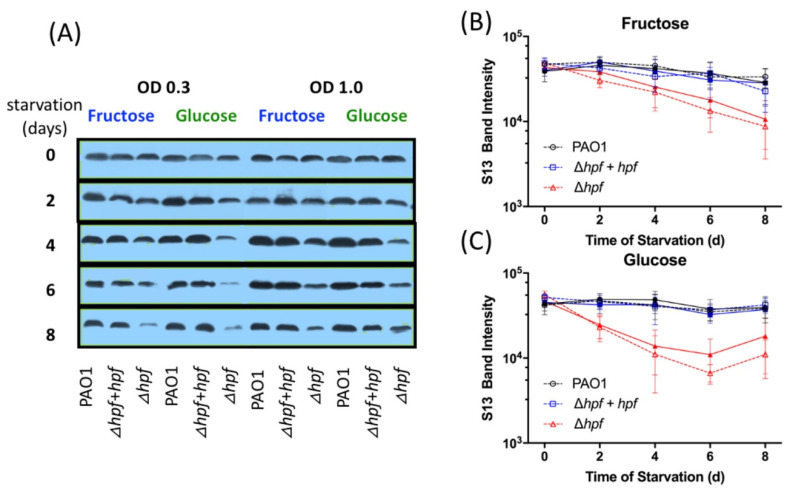
Quantification of *P. aeruginosa* S13 band intensity over the course of 8 d of starvation, from whole cell extracts. Strains were pre-cultured in MOPS minimal media with either fructose or glucose as the sole carbon source. (**A**) Representative immunoblot of S13 from whole cell extracts. (**B**) Quantification of S13 signal intensity obtained by densitometry of three immunoblots, when strains were cultured in MOPS-fructose. Solid lines and symbols indicate cells were pre-cultured to O.D._600_ = 1.0 and dashed lines and open symbols indicate cells were pre-cultured to O.D._600_ = 0.3. (**C**) Quantification of S13 signal intensity when strains were cultured in MOPS- glucose.. Data show the mean and standard error of three independent biological replicates.

**Figure 10 ijms-21-09494-f010:**
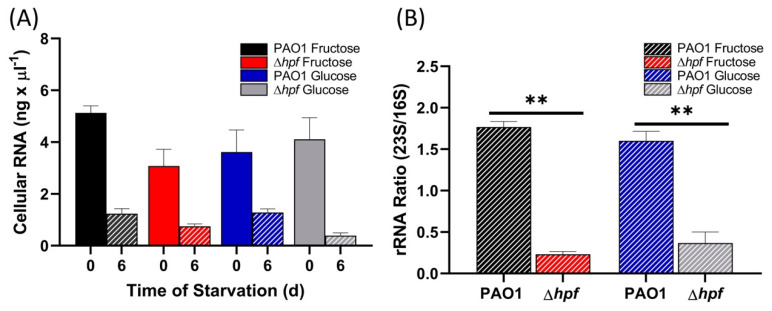
RNA integrity of *P. aeruginosa* cultured in MOPS-glucose or MOPS-fructose to O.D._600_ = 1.0, then starved for six days. (**A**) Total cellular RNA levels prior to starvation and after 6 d of starvation. (**B**) 23S/16S rRNA ratios of strains pre-cultured in MOPS-glucose or MOPS-fructose, then starved for 6 d. Data show the mean and standard error of three independent biological replicates. Asterisks (**) for 23S/16S ratio shows significant difference between 0 and 6 d (*p* < 0.001).

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
