# Peer review of "Role of Hibernation Promoting Factor in Ribosomal Protein Stability during Pseudomonas aeruginosa Dormancy"

_ijms, 2020, doi:10.3390/ijms21249494_

Round 1

Reviewer 1 Report

The ms by Theng et al analyzes the impact that the HBF protein plays for the stability of ribosomal proteins in Pseudomonas aeruginosa. This study is a follow up study from a previous publication of the group. The ms is well written and  the study appears to be carefully designed. I recommend publication after the following small critiques have been addressed.

1) The file with the supplemental data contains the figures of the main text. Therefore, I was unable to evaluate all supplemental data.

2) Fig 1A & B. Can this be quantified by densitometric analysis?

3) It is interesting that the hpf deletion strain shows fewer CFU but equal protein abundance. How were the strains normalized. This information should be added to the material and methods.

4) Some figures are overloaded w/ data and could presented more clearly, e.g. Figure 3. Why not separating the 1.0 and 6.0 OD so that the reader can better see the differences.

5) What is the rationale for choosing Fructose as alternative carbon source.

Author Response

Authors’ responses to reviewer comments are in bold:

Reviewer 1:

The ms by Theng et al analyzes the impact that the HBF protein plays for the stability of ribosomal proteins in Pseudomonas aeruginosa. This study is a follow up study from a previous publication of the group. The ms is well written and  the study appears to be carefully designed. I recommend publication after the following small critiques have been addressed.

1) The file with the supplemental data contains the figures of the main text. Therefore, I was unable to evaluate all supplemental data.

The supplemental file was re-uploaded and should be correct now.

2) Fig 1A & B. Can this be quantified by densitometric analysis?

The bands can be quantified by densitometry.  However, the results in Fig 1 are qualitative.  So, we don’t believe that quantifying those bands will provide new information.  In fact, we previously determined total protein amounts and total rRNA amounts for those samples but decided not to include those figures in the final manuscript, since there may be variation in the ribosome preps that complicate quantification.  The quantitative information begins with Fig 3.

3) It is interesting that the hpf deletion strain shows fewer CFU but equal protein abundance. How were the strains normalized. This information should be added to the material and methods.

The strains were normalized by adjusting all of the initial starting cultures to approximately 2 x 108 cells/ml.  We’ve added this information to the materials and methods (lines 500-515)

4) Some figures are overloaded w/ data and could presented more clearly, e.g. Figure 3. Why not separating the 1.0 and 6.0 OD so that the reader can better see the differences.

This is a good suggestion.  Figure 3 does have a lot of information.  However, showing both O.D. = 1 and O.D. = 6 adjacent to each other allows the reader to make a direct comparison between the different conditions.

5) What is the rationale for choosing Fructose as alternative carbon source.

In preliminary experiments we tried several different carbon sources.  We chose to compare glucose and fructose because those two carbon sources gave distinctive differences in growth and hpf expression.  We’ve added text to the section from 253-259 to describe the rationale for using glucose and fructose. 

Reviewer 2 Report

In actively growing bacteria, their ribosomes constantly synthesize new proteins to maintain vital functions. However, under certain conditions, especially when there are not enough nutrients, bacteria slow down protein synthesis and convert their ribosomes into inactive yet stable forms. This process of inactivation is called “ribosome hibernation” and is mediated by the so-called hibernation factors. HPF is one of these factors playing a crucial role in hibernation. Besides the fundamental significance of this process, it is also linked to acquired drug resistance in pathogenic bacteria. More than half of clinically used antibiotics directly target the actively working ribosomes, but obviously, the same antibiotics are useless in targeting inactive hibernating ribosomes. Therefore, just by making their ribosomes inactive, pathogenic bacteria can escape the action of antibiotics, ultimately leading to drug resistance. Previously, it was shown that during the stationary phase, HPF binds to ribosomes and converts them into inactive 100S dimer form. Importantly, in such complexes, ribosomal RNA is protected against degradation in starved cells. In the presented work by Theng et al., the authors study the role of the hibernation factor in stabilizing of rProteins during nutrient limitation. They found that two ribosomal proteins L5 and S13 are not stable and actively degraded in the delta_hpf cells throughout starvation. Thus, HPF is required to protect rRNA and rProteins against degradation during nutrient deprivation. Interestingly, the authors show that the stringent response system (relA and spoT) is not involved in such protection but prevents cellular proteome from proteolytic degradation and cell lysis. In my opinion, overall, this study adds important evidence for the role of HPF in bacterial hibernation.

I would also like to emphasize that the presented study probably uncovers a function of HPF beyond the stabilization of fully assembled ribosomes in starved bacterial cells. The fact is that the ribosomes and ribosomal RNAs are degraded to some significant extent, even in the presence of HPF (figure 1 and black bars on figure 4A). In contrast, proteins L5 and S13 stay stable during prolonged starvation in wild-type cells and degraded only in the absence of the hibernation factor (figure 3). This can potentially point to the HPF-mediated protection of ribosome-free ribosomal proteins that are released from degraded ribosomes. If this is the case, such protection can explain why dormant P. aeruginosa cells can efficiently survive severe nutrient stress. The cells can probably restore the concentration of active ribosomes after starvation exclusively due to the transcription of the 16S, 23S and 5S rRNA followed by assembly of nascent transcripts with HPF-stabilized preexisted ribosomal proteins. The authors might discuss this possible role of the hibernation factor in the manuscript.

In my opinion, the results of this study will be interesting for readers of the journal, and the presented manuscript is acceptable for publication. The authors should address only several minor points.

  1. The authors show that two ribosomal proteins L5 and S13 are degraded in the absence of HPF during starvation. In this regard, since ribosomal proteins represent a significant fraction of the total cellular protein, it looks strange that the cellular protein remains constant throughout starvation regardless of HPF (figure 2B). How does the spectrum of cellular proteins look like on SDS-PAGE after starvation with and without HPF?
  2. Figure 4B shows that the 23S/16S ration is significantly decreased in the delta_hpf cells, probably due to preferential degradation of 23S rRNA. However, from biochemical and structural data that HPF interacts with the small ribosomal subunit. Hence, the expectation is that the HPF absence should affect first of all stability of 16S rather than 23S rRNA. How can the authors explain this?

Author Response

Reviewer 2: Comments and Suggestions for Authors

Author’s response in bold:

In actively growing bacteria, their ribosomes constantly synthesize new proteins to maintain vital functions. However, under certain conditions, especially when there are not enough nutrients, bacteria slow down protein synthesis and convert their ribosomes into inactive yet stable forms. This process of inactivation is called “ribosome hibernation” and is mediated by the so-called hibernation factors. HPF is one of these factors playing a crucial role in hibernation. Besides the fundamental significance of this process, it is also linked to acquired drug resistance in pathogenic bacteria. More than half of clinically used antibiotics directly target the actively working ribosomes, but obviously, the same antibiotics are useless in targeting inactive hibernating ribosomes. Therefore, just by making their ribosomes inactive, pathogenic bacteria can escape the action of antibiotics, ultimately leading to drug resistance. Previously, it was shown that during the stationary phase, HPF binds to ribosomes and converts them into inactive 100S dimer form. Importantly, in such complexes, ribosomal RNA is protected against degradation in starved cells. In the presented work by Theng et al., the authors study the role of the hibernation factor in stabilizing of rProteins during nutrient limitation. They found that two ribosomal proteins L5 and S13 are not stable and actively degraded in the delta_hpf cells throughout starvation. Thus, HPF is required to protect rRNA and rProteins against degradation during nutrient deprivation. Interestingly, the authors show that the stringent response system (relA and spoT) is not involved in such protection but prevents cellular proteome from proteolytic degradation and cell lysis. In my opinion, overall, this study adds important evidence for the role of HPF in bacterial hibernation.

I would also like to emphasize that the presented study probably uncovers a function of HPF beyond the stabilization of fully assembled ribosomes in starved bacterial cells. The fact is that the ribosomes and ribosomal RNAs are degraded to some significant extent, even in the presence of HPF (figure 1 and black bars on figure 4A). In contrast, proteins L5 and S13 stay stable during prolonged starvation in wild-type cells and degraded only in the absence of the hibernation factor (figure 3). This can potentially point to the HPF-mediated protection of ribosome-free ribosomal proteins that are released from degraded ribosomes. If this is the case, such protection can explain why dormant P. aeruginosa cells can efficiently survive severe nutrient stress. The cells can probably restore the concentration of active ribosomes after starvation exclusively due to the transcription of the 16S, 23S and 5S rRNA followed by assembly of nascent transcripts with HPF-stabilized preexisted ribosomal proteins. The authors might discuss this possible role of the hibernation factor in the manuscript.

This is a very good point.  Certainly, some free ribosomal proteins are maintained in the as “back-ups” to replace damaged proteins, as described in reference 50.  While the reviewer is probably correct on this point, at present, we do not have a way of distinguishing ribosome-associated rProteins from the free rProteins.

In my opinion, the results of this study will be interesting for readers of the journal, and the presented manuscript is acceptable for publication. The authors should address only several minor points.

  1. The authors show that two ribosomal proteins L5 and S13 are degraded in the absence of HPF during starvation. In this regard, since ribosomal proteins represent a significant fraction of the total cellular protein, it looks strange that the cellular protein remains constant throughout starvation regardless of HPF (figure 2B). How does the spectrum of cellular proteins look like on SDS-PAGE after starvation with and without HPF?
  2. Figure 4B shows that the 23S/16S ration is significantly decreased in the delta_hpf cells, probably due to preferential degradation of 23S rRNA. However, from biochemical and structural data that HPF interacts with the small ribosomal subunit. Hence, the expectation is that the HPF absence should affect first of all stability of 16S rather than 23S rRNA. How can the authors explain this?

Both of these points are excellent and have been the topics of discussions in our lab meetings: 

Point 1 – why is there a decrease in ribosomal proteins but not in total protein? 

I think that this is due to the resolution of the protein assay technique.  If the ribosomes make up 25-50% of the total cell protein at the initiation of starvation, then a 25-50% decrease in total cell protein may be within experimental error of the protein assay technique.  We see a slight reduction in total protein over the course of starvation in several Figure 2, but the decrease is not statistically significant.  Also, for the other pre-growth conditions, we see an approximate 2-fold decrease in total protein (Fig S4, S5)., which would be expected if ribosomal proteins are lost.  It is likely that the decrease in total protein is within the standard deviation of the measurements.  (We have run SDS-PAGE gels and stained with coomassie blue.  We didn’t include that experiment in this manuscript, but banding pattern looks similar through starvation).

Point 2 – Why is there a selective loss of the 23S rRNA rather than the 16S rRNA? 

This is another interesting question, and I don’t have a good explaination.  I agree with the reviewer that based on the way that rRNA is cleaved and degraded the 16S rRNA should be lost first (reference 48).  However, the reduction in 23S/16S ratio in the Dhpf mutant is highly reproducible.  We show this phenomenon in references 21 and 22.  In reference 21, in addition to the 23S/16S rRNA ratio, we performed fluorescence in situ hybridization (FISH) on the 16S rRNA, and show that in fact, both the 16S and the 23S rRNAs are lost in the Dhpf mutant.  However, for reasons that we can’t explain yet, some of the 16S rRNA seems to remain intact longer than the 23S.  Therefore, the change in 23S/16S rRNA ratio is an excellent indicator of ribosome integrity.